# Effects of Fall Training Program on Automatization of Safe Motor Responses during Backwards Falls in School-Age Children

**DOI:** 10.3390/ijerph16214078

**Published:** 2019-10-23

**Authors:** Óscar DelCastillo-Andrés, Luis Toronjo-Hornillo, Luis Toronjo-Urquiza

**Affiliations:** 1Departamento de Educación Física y Deporte, Universidad de Sevilla, C/Pirotecnia s/n, 41013 Sevilla, Spain; ltoronjo@gmail.com; 2Department of Chemical and Biological Engineering, The University of Sheffield, Sheffield S10 2TN, UK; ltoronjourquiza1@sheffield.ac.uk

**Keywords:** childhood injuries, public health, health promotion, protective factors

## Abstract

A significant number of children suffer injuries from falls. The current measures of prevention and education regarding falls are not sufficient, as falling is still the main cause of injury at this age. This study aimed to evaluate the risk of injury during a backward fall and assess the effect of the Safe Fall training program on this risk. 457 primary school children between the ages of 6 and 12 (mean age of 9) were enrolled in a 6-week randomized intervention. The program was carried out during physical education classes and consisted of an intervention group that followed the Safe Fall training program and a control group that was given equilibrium exercises. The risk of injury was assessed before and after the implementation using the Information Scale on Safe Ways of Falling observation scale, evaluating the responses of five different body parts (head, hip, knees, upper limbs and back). Students’ natural response to falls was associated with a high risk of injury in more than 90% of the cases. The implementation of the Safe Fall program resulted in a considerable decrease in this risk, with percentages lowered to levels between 8.7% and 18.3%.

## 1. Introduction

Unintentional falls are the leading cause of non-fatal injuries in the pediatric population [1]. This is due to the fact that children do not have fully developed reflexive and balancing abilities, as the motor skills which empower balance capacity mainly develop between the ages of 5 and 10 [2,3]. It is estimated that the healthcare expenses arising from fall-related injuries in children in the United States are approximately $58 billion per year [4]. This, together with the fact that children who experience medically attended injuries have a higher risk of recurrence, stresses the need for further interventions [5]. 

Measures to prevent falls and fall-related injuries have been promoted in many countries for over 10 years. Nevertheless, the number of falls in children has increased and there has been no change in the incidence of events such as major trauma [6]. Furthermore, existing measures focus solely on prevention, not on mitigating the outcomes of an unintentional fall once it takes place [7,8]. Although physical education (PE) classes can improve balance, they do not teach children to modify their natural motor responses to render them safer in the event of a fall [9,10]. Only one program attempted to tackle fall injuries by teaching children how to fall; the results were not conclusive [11].

Taking into account the considerable effects that fall-related injuries have on quality of life and the costs that they entail, more needs to be done. The aim of this study was to assess the natural response of children to a sudden fall and to evaluate whether the implementation of a falls training program (Safe Fall) would result in a reduced risk of injury. This program is based on fundamental judo falling techniques. These techniques have been isolated, further developed and can be implemented in PE classes. The goal of the Safe Fall program is to make children more self-sufficient in case of a sudden fall, lowering the number of injuries, their severity and their consequences. 

## 2. Materials and Methods

### 2.1. Sample

The population consisted of primary school students attending the Jacaranda Primary School Center in Seville (Spain). Primary schools in Spain have six academic years and the study included three classes from each year. On average each class contained 31 students, amounting to a sample of 459 children. Students with serious learning and developmental disabilities, such as Down syndrome or autism, participated in the program, but their responses were not included in the analysis. 

### 2.2. Structure

The project was a randomized control trial and was performed between January and April 2018, in two six-week cycles. During the first cycle, two randomly chosen classes from each academic year (the intervention group) followed the Safe Fall program, while a third class formed the control group and practiced equilibrium exercises. During the second cycle, the classes forming the control group in the first cycle were swapped to the intervention arm and followed the Safe Fall program. The initial intervention group completed the program at the end of the first cycle.

### 2.3. Blinding

The students were blinded to the intervention allocation. All the schoolteachers were fully aware of the purpose of the program and the intervention protocol. The external observer who rated the risk associated with the fall was blinded to the intervention allocation. 

### 2.4. Ethics Approval

The research complied with the approval of the Ethics Committee of the Biomedical Research of Andalusia and was focused on ensuring the safety of the child throughout the execution process. Likewise, the research complied with the approval of the Ethics Committee of the Biomedical Research of Andalusia and was focused on ensuring the safety of the child throughout the execution process (CI-0021-N-18). 

### 2.5. Implementation of the Safe Fall Program and Assessments

The assessment included a baseline test to assess the children’s natural response to an unexpected backwards fall. The risk of injury was assessed using the Information Scale on Safe Ways of Falling observation scale, which identifies potentially harmful responses (PHR) during a fall [12]. During the test, the participants were placed in a squatting position on top of a polyurethane foam mattress by an assistant holding their arms. The assistant asked them to close their eyes and brought them into a position of imbalance. The students were then released to fall backward on the mattress. The responses to the fall of five body parts (head, back, upper limbs, hips and knees) were assessed and scored [12]. The following gestures and positions were considered potentially harmful responses (PHR): head—lack of capacity to keep the head under control and close to the chin; back—landing on a flat back; upper limbs—extension towards the floor to stop the fall; hips—lack of bending towards the torso; knees—lack of flexing, landing with straight legs (Figure 1). Each of the five body parts was scored independently. A score of 1 was given in the event of a PHR from that part and a score of 0 was given if there was no PHR observed. This led to an overall score between 0 and 5, with higher scores suggesting higher risk of injury. All the tests were filmed and stored for future reference.

In this stage, a series of basic student characteristics were measured, which were correlated to the results obtained in the tests. This served to understand if these characteristics influenced the effectivity of the interventions of the Safe Fall program. Height and weight were measured and information on the children’s sport participation was recorded. The latter variable was defined as the amount of sports activity outside school hours and was categorized as low, moderate or high. ‘Low sport participation’ indicated the lack of any sports activities outside school. ‘Moderate sport participation’ was defined as practicing sports activities up to 3 times a week or up to a total of 3 h per week. ‘High sport participation’ was anything exceeding moderate participation in number of sessions or total time. This information was gathered from the interviews with participants and was combined with the information gathered from parents and teachers.

Students’ PE marks obtained in the regular PE program were also recorded. These are given on a 1–10 scale at the PE teacher’s discretion, based on the evolution of the student’s motor skills over the academic course, as well as their understanding of theoretical concepts and attitude.

The program was implemented using linear pedagogy [13], focusing on the assimilation and subsequent automatization of safe motor responses. The program was then implemented in three phases, as part of PE classes. The first part was a 60-min presentation about falls and their consequences at school age, showing safe and unsafe positions in case of a fall. A large part of this first session consisted of exercises and games, to allow students to put the theory into practice and start assimilating the safe motor responses.

The 10 successive 10-min sessions took place during the warm-up phase of PE classes. They consisted of two 5-min exercises or one 10-min exercise. These exercises followed a sequence based on their difficulty. The teacher would review some known exercises, adapt them when necessary and introduce new ones depending on the level of students’ achievements. In the starting exercises of the sequence, the students’ center of gravity was on the ground. The progression first focuses on increasing mobility while doing the exercises, then speed of execution is increased, and final exercises raise the center of gravity to a position in which the students’ knees are bent at a 90-degree angle. This limitation of the height of the center of gravity helps avoiding injuries among the minors involved in the program.

The control group program followed the exact same structure (a 1-h initial session followed by 10 10-min sessions) as the intervention group, with the difference being that the theoretical and practical contents focused on preventing falls through balance/imbalance exercises. These contents were chosen because they allowed the entire school to follow a fall injury prevention program. Thus, the children within the control group were doing the same amount of physical exercise focusing on the same subject matter, but, crucially, were not working on motor responses that occur during a fall.

The implementation of the program during PE classes was carried out by several teachers from the same school. These teachers received theoretical and practical instructions for the execution of the program, which consisted of: the theory of safe and non-safe motor responses associated with falls, World Health Organization data describing the impact of falls on the health of school children, theoretical and practical training on data collection processes and tools (including example videos of subjects performing the test to make sure teachers scored consistently between them), as well as the exercises for each control and intervention session. Each teacher was assigned both intervention and control groups.

Finally, a post-test identical to the baseline test was carried out. The same structure was followed in the control group. However, these students practiced equilibrium exercises instead of safe fall exercises. Students in the control group were tested for a third time after changing groups and completing the Safe Fall program (Figure 2). 

### 2.6. Statistical Analysis

The chi-squared test was used to compare the categorical baseline characteristics, while the student’s *t*-test and the Mann-Whitney test were used for continuous variables. The number of students showing PHR for each body part was assessed and compared between the two groups using the chi-squared test. Kruskal-Wallis was used to compare the PHR score between tests for the intervention and control arms. The variables studied were the allocation arm, gender, academic year, PE mark (on a scale from 1 to 10, 10 being the highest), body mass index (BMI) and sport participation. In order to assess the influence of the variables on the outcomes, ANCOVA analysis was performed. *p*-values below 0.05 were considered significant. Data was analyzed using SPSS V 25.0 (SPSS Inc., Chicago, IL, USA).

## 3. Results

### 3.1. Baseline Characteristics

The study included 457 students aged 6–12 years, approximately half of them female. From these, four were excluded from the analysis due to learning disabilities and three because they could not complete the training.

Sixty percent of the population fell in the ‘low sport participation’ category, while only 10% were highly involved in sports. Baseline characteristics were similar between the two groups (Table 1).

When studying the head, neck and upper limbs, a similar number of students showed potentially harmful responses at baseline, with no statistical differences between the groups (i.e., both the control group and the intervention group had similar risks of injury at baseline). There was a statistical difference for hip and knee responses, with fewer students showing PHR in the intervention arm (Figure 3). This was reflected in the PHR score at baseline, with the control group scoring higher than the intervention group (mean scores of 4.86 and 4.64, respectively). However, when studying the relative risks, the confidence intervals at baseline included 1 for all the body parts studied, showing that there was no statistical difference in the two groups at baseline, i.e., they had similar risks of injury for all body parts (Table 2).

### 3.2. Effect of the Safe Fall Program on PHR

The implementation of the Safe Fall program in the intervention group caused a significant decrease in the percentage of students showing PHR. This was observed for all the body parts studied. The part affected the most was the upper limb, with an 88.7% decrease (Figure 3).

The implementation of the equilibrium exercises in the control group slightly improved the response of students, but the decrease of the number of students showing PHR was not significant. A maximum drop of 3.3% was observed for the knees. The percentage of PHR remained above 90% for all the parts studied (Figure 3).

The average PHR score remained high in the control group between the baseline score of 4.86 (4.78–4.94 95% CI) and the score at the end of the control program of 4.78 (4.64–4.91 95% CI). The control group baseline score also did not differ significantly from the intervention group’s baseline score of 4.65 (4.54–4.76 95% CI). When the intervention group (and, later, the control group) completed the intervention program, their average PHR scores decreased significantly. In the final test, the intervention group scored a 0.79 (0.62–0.96 95% CI) while the final test for the control group resulted in a final score of 0.41 (0.27–0.56 95% CI) with no significant differences between them (Figure 4).

### 3.3. Effects of Different Variables on the Outcomes

In order to assess the influence of variables on the PHR score a covariate study was performed on the baseline test and final test. The analysis performed at baseline showed that gender, sport participation and the allocation arm were not associated with the PHR score (*p* > 0.05). On the other hand, the academic year, BMI and PE mark were all significantly related (Table 2).

When looking at the trend of those variables in more detail, the PHR score tended to decrease as students grew older, dropping from a mean of 4.81 in the first year, to a mean of 4.47 in the last year. Children in their third and fourth year had higher PHR scores, indicating higher risk of injury (Figure 5A). Children with higher PE marks had lower PHR scores, indicating a reduced risk of injury (Figure 5B). Obese (OB) individuals had higher average PHR scores at baseline compared with normal weight (NW) individuals (Figure 5C). On the other hand, the PHR score varied significantly in the underweight students.

When the same analysis was performed for the final test, none of the variables (gender, academic year, sport participation, BMI, PE mark) turned out to be significant. The only variable that was found to be significant was the allocation arm, indicating that all students could benefit from this intervention, independent of their baseline characteristics.

## 4. Discussion

The present study has shown that children aged six to twelve have a high risk of injury when falling backwards. Their natural, spontaneous responses are potentially harmful and are related with injuries in several body parts like the head, hips, upper and lower limbs [14,15,16]. However, the implementation of the Safe Fall program can mitigate this risk significantly by correcting these potentially harmful responses.

The introduction of the Safe Fall program significantly reduced the PHR scores in all children, with the effect being independent of gender, academic year, sport participation, BMI, and PE mark. Although equilibrium exercises were found to be useful in previous studies [9], the results of this study showed that they had no significant benefit in reducing the risk of injury once the fall occurs.

Although PE classes are fundamental for the development of children, the activities and sports involved do not tend to have an impact on the children’s response to a sudden fall. Previous studies identified that practicing sports did not lower the risk of suffering an injury, except when the sport was associated with falling mechanics [9].

Other studies had investigated the mechanics of falls, identifying techniques to reduce the impact when falling. These involved the flexion of extremities, rolling over, hitting the ground with an open hand, relaxing the muscles and staggering the impact [17]. These techniques contributed to the reduction of the impact of the fall and are congruous with the parameters measured and taught during this study.

To our knowledge, there is only one previous study that assessed the problem of falls in children by teaching falling techniques in school populations [11]. The study showed that the change in fall injury rates after the intervention was not significant; the only exception was in less active kids. The present report found no differences in the learning behaviour for children with low sport participation, which indicates that it is an efficient way to lower the risk of injury in a group of children which are more susceptible to injuries during physical activity [18].

Different strategies, such as prevention mechanisms and educating the population, have been implemented in order to address the issue of fall-related injuries [19,20]. An integrated program on the risk of falling which involved children, their family and their school was carried out in rural China. This program managed to reduce the number of falling events by half [21]. Other studies have suggested that parental awareness of the risk of injury is a short-term strategy and diminishes within 6 months after a fall [5]. These results are thought to be due to the fact that parents dissociate the injury protection behaviour from their children’s risk engagement [22]. In contrast, continuous information and sensibilization campaigns regarding the risk of falls, although efficient at delivering the message, are inconsistent in lowering the number of fall-related injuries [23].

## 5. Limitations

This study had several limitations. The implementation and collection of the data was carried out over two 6-week periods, so the long-term effects of this program could not be evaluated. Implementation of this program during the entire length of the academic year could include more elements, such as falls in other directions. Furthermore, this study was done in a single school, and the transferability of the context to other schools and environments has not been addressed. More research needs to be carried out in this field, as there is only a limited number of studies into the effect of programs that improve fall-related motor skills. There are a lot of studies that focus on educating the population or developing fall prevention programs; however, they do not allow a direct comparison with this study.

The intervention and control groups of this study did not show any significant differences in the general baseline characteristics. Nevertheless, a significant difference was observed in the number of students showing PHR in the hips and knees when comparing the control and intervention groups. This could be a result of randomizing by class or the effect of other variables not considered in this study.

## 6. Conclusions

Children at school age have natural motor responses to backward falls associated with a high risk of injury. The implementation of the Safe Fall program was found to significantly reduce the risk of injury in all children, independently of gender, BMI, PE grade, academic year and level of sport participation.

## Figures and Tables

**Figure 1 ijerph-16-04078-f001:**
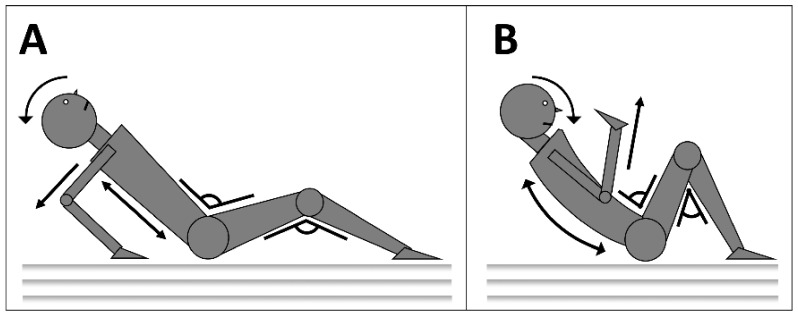
The figure illustrates the potentially harmful responses (PHR) (**A**) and protective responses (**B**) of the five body parts (head, back, upper limbs, hips and knees) assessed during a backward fall.

**Figure 2 ijerph-16-04078-f002:**
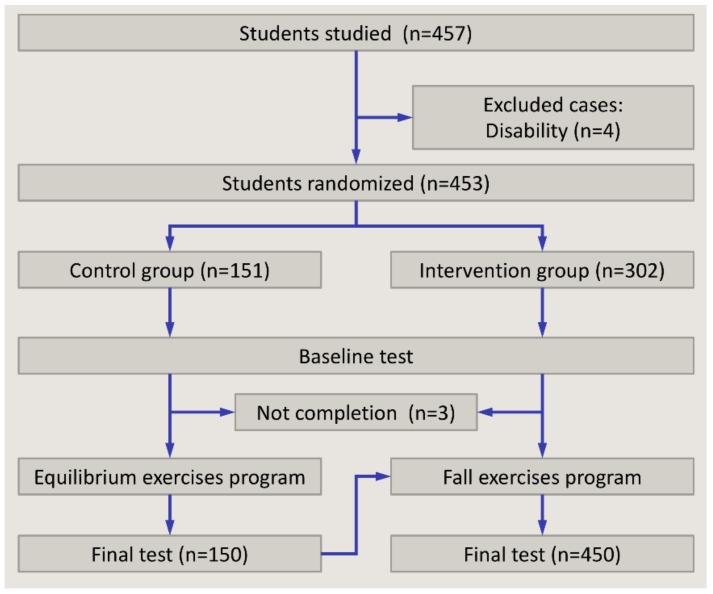
Students’ flow through study.

**Figure 3 ijerph-16-04078-f003:**
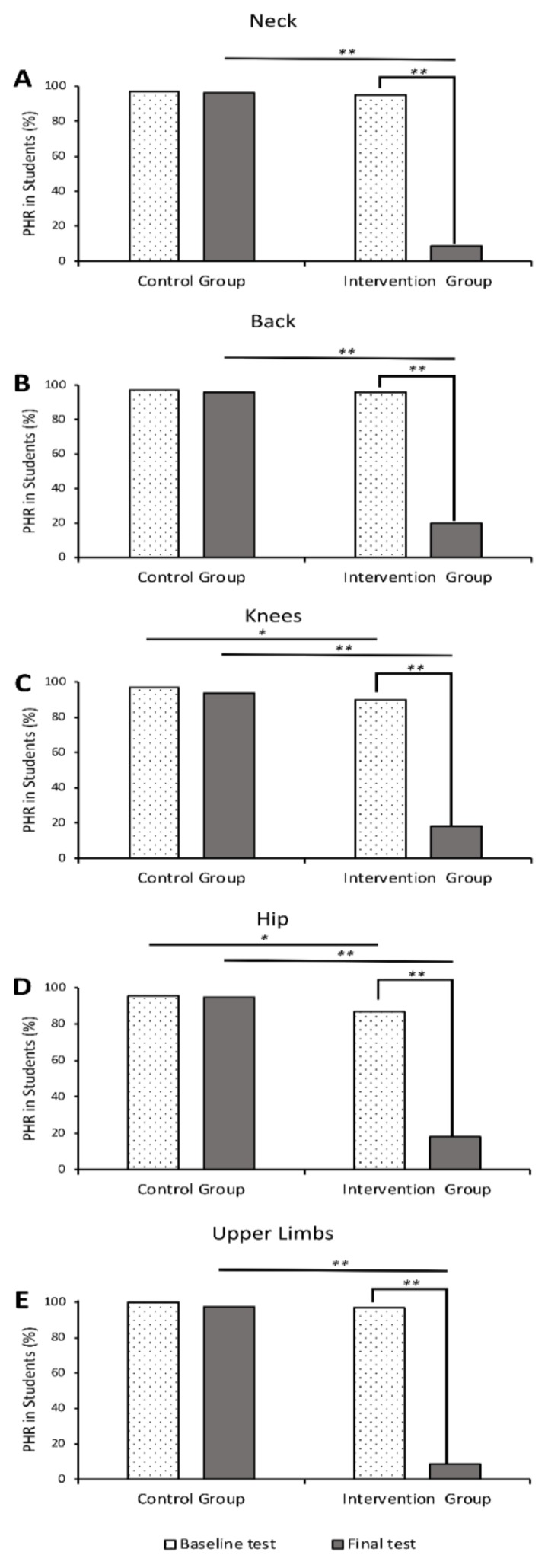
Graphs showing the percentage of students showing potentially harmful responses (PHR) for the different body parts studied: neck (**A**), back (**B**), knees (**C**), hips (**D**) and upper limbs (**E**). The percentages are shown both for the control and intervention groups. The results for the baseline test are illustrated in the dotted bars, while the results for the final test are shown in the solid grey bars. A chi-squared test was used to evaluate statistical differences. *p*-values were reported when <0.05 (*) and <0.01 (**).

**Figure 4 ijerph-16-04078-f004:**
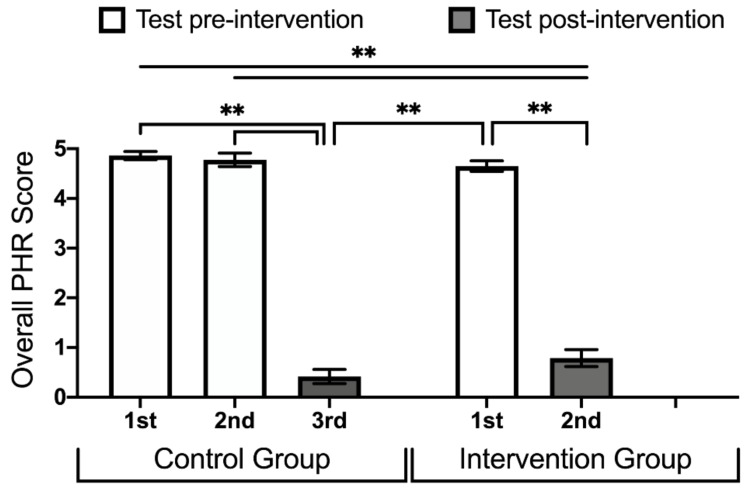
Average of the potential harmful response (PHR) score for each test taken on the intervention arm (two tests) and the control arm (three tests). White bars represent tests at baseline or after the implementation of the control program. Grey bars represent the results from the test after the intervention program. A Kruskal-Wallis test was used to evaluate statistical differences. *p*-values were reported when <0.05 (*) and <0.01 (**).

**Figure 5 ijerph-16-04078-f005:**
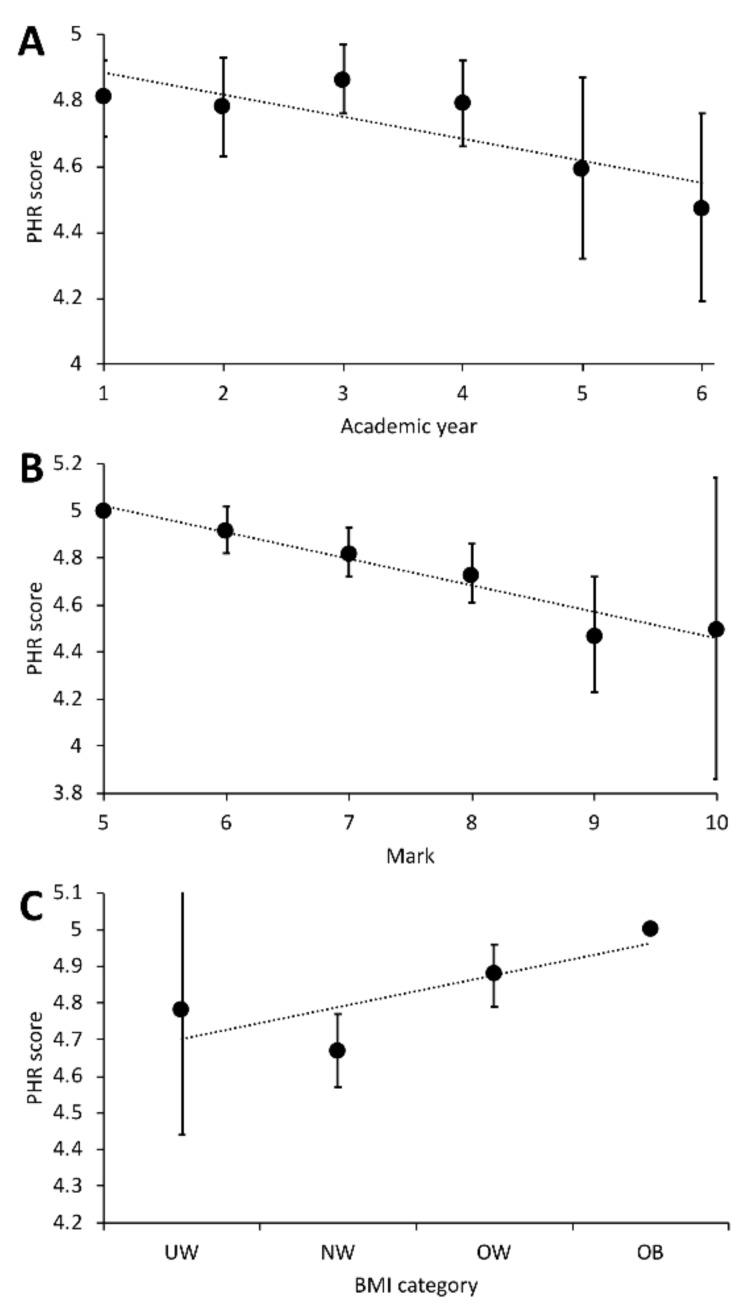
Students’ average potentially harmful response (PHR) score at baseline by academic year (**A**), mark (**B**) and body mass index (BMI) (**C**). BMI categories: underweight (UW), normal weight (NW), overweight (OW) and obese (OB). Error bars represent the 95% confidence intervals.

**Table 1 ijerph-16-04078-t001:** Baseline characteristics of the two groups.

	Control	Intervention	*p*-Value
Participants ^3^	151 (33)	302 (67)	
Sex (Females) ^3^	77 (51.0)	151 (50.0)	0.921 ^a^
Academic year ^3^			0.955 ^b^
1st	25 (16.6)	49 (16.2)	
2nd	26 (17.2)	52 (17.2)	
3rd	26 (17.2)	50 (16.6)	
4th	24 (15.9)	53 (17.5)	
5th	25 (16.6)	50 (16.6)	
6th	26 (17.2)	48 (15.9)	
PE mark ^1^	7.72 (1.00)	7.77 (1.06)	0.529 ^c^
BMI (kg/m^2^) ^2^	18.71 (18.15, 19.28)	18.44 (18.02, 18.86)	0.252 ^b^
Categories ^3^			0.711 ^d^
Underweight	4 (2.7)	6 (2.0)	
Normal	114 (76.0)	232 (77.3)	
Overweight	29 (19.3)	51 (17.0)	
Obese	3 (2)	11 (3.7)	
Sport participation ^3^			0.196 ^a^
Low	102 (67.5)	180 (59.8)	
Moderate	32 (21.2)	87 (28.9)	
High	16 (10.6)	34 (11.3)	

Note. ^1^ Mean (standard deviation); ^2^ mean (IC range); ^3^ count (percentage); ^a^ chi-squared; ^b^ Mann Whitney U Test; ^c^ independent *t*-test; ^d^ Fisher’s exact test; physical education; body mass index.

**Table 2 ijerph-16-04078-t002:** Effect of variables on the PHR score at baseline and final test.

Variables	Baseline Test	Final Test
Sex	0.739	0.296
Academic year	0.002	0.145
PE mark	0.025	0.747
BMI	0.015	0.233
Sport participation	0.617	0.607
Intervention	0.153	<0.001
Baseline	---	0.055

*p*-values of variables by covariate study at baseline and final test; (PHR) potential harmful responses; (BMI) body mass index; (PE) physical education.

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
