# Peer review of "Effects of Fall Training Program on Automatization of Safe Motor Responses during Backwards Falls in School-Age Children"

_ijerph, 2019, doi:10.3390/ijerph16214078_

Round 1
Reviewer 1 Report
The purpose of this study is to assess whether a Safe Fall training program reduces the risk of injury following backward fall of school-age children. In our opinion the study is well done.
The sample involved in the study is very broad and this is a strong point.
We suggest to present the details of the program, then accurately describe which are the safe fall exercises that were used in the program, how many times they were repeated and the didactic progression during the six-week cycles, but above all specify which type of didactic approach has been used (for example linear, non linear pedagogy and why)
The details must be described also for the balance exercises made by the control group. It is important to know why the authors decided to have a balance program followed by the control group and if contextual interference in two groups' program have been considered.
It would be useful to know if the teacher was always the same. In case there were different teachers if they were trained and which were the instructions provided to them.
Some things should be fixed regarding how the data was presented in the tables.
In table 1 there are some notes without footer explanation, while in the caption there are symbols that are not found in the table.
Physical education marks are not properly specified: it must be described which assessments have been made, if based on specific sport ability, coordination, agility, or other stuff.
Table 2 compares initial and final tests results for the two groups, but we think it is not clear how they are presented and what are the values in brackets.
Author Response
Answer to the Reviewer’s comments
Special Issue " Health Promotion and Quality of Life Improvement Throughout the Life Cycle”
of International Journal of Environmental Research and Public Health
Manuscript ID: ijerph-598128
Dear Editor,
Enclosed you will find a revision of our manuscript: Effects of fall training program on automatization of safe motor responses during backwards falls in school-age children. We would like to thank the Editor for giving us the opportunity to resubmit our work and to the reviewers for their thoughtful and constructive comments. We have considered all the suggestions and have incorporated them into the revised manuscript. An itemized point-by-point response to the reviewer’s comments is presented below.
Comments to the Author
Reviewer: 1
The purpose of this study is to assess whether a Safe Fall training program reduces the risk of injury following backward fall of school-age children. In our opinion the study is well done.
The sample involved in the study is very broad and this is a strong point.
1.- We suggest to present the details of the program, then accurately describe which are the safe fall exercises that were used in the program, how many times they were repeated and the didactic progression during the six-week cycles, but above all specify which type of didactic approach has been used (for example linear, non linear pedagogy and why).
We appreciate the comment of the reviewer, and we add the following text in response:
page 2 and 3, lines 79-93.
“In order to implement the program, linear pedagogy was used [1], focusing on the assimilation and subsequent automatization of safe motor responses. The program was then implemented in three phases, as part of P.E. classes. The first part was a 60-minute presentation about falls and their consequences at school age, showing safe and unsafe positions in case of a fall. A large part of this first session consisted of exercises and games, to allow students to put the theory into practice and start assimilating the safe motor responses.
The 10 successive 10-minute sessions took place during the warm-up of the physical education class. They consisted of two 5-minute exercises or one 10-minute exercise. These exercises, which followed a sequence depending on their level of difficulty, were adjusted by the teacher to the learning progressions established in the program. The teacher would review some known exercises and introduce new ones depending on the level of students’ achievements. In the starting exercises of the sequence, the centre of gravity was on the ground. The progression first focuses on increasing mobility while doing the exercises, then speed of execution is increased, and final exercises raise the centre of gravity to a position in which in which the knees are bent at a 90-degree angle. This limit helps avoiding injuries among the minors involved in the program.”
2.- The details must be described also for the balance exercises made by the control group. It is important to know why the authors decided to have a balance program followed by the control group and if contextual interference in two groups' program have been considered.
We will like to thank the reviewer for point out this to us; to provide the information requested we included the following section at page 3, lines 94-99.
“The control group program followed the exact same structure (1-hour initial session followed by 10 10-minute sessions) as the intervention group, with the difference that the theoretical and practical contents focused on preventing falls through balance/imbalance exercises. These contents were chosen because they allowed the entire school to follow a fall injury prevention program [2]. Thus, the children within the control group were doing the same amount of physical exercise focusing on the same subject matter, but, crucially, were not working on motor responses that occur during a fall.”
3.- It would be useful to know if the teacher was always the same. In case there were different teachers if they were trained and which were the instructions provided to them.
We included the following information as requested on page 3, lines 100-107.
“The implementation of the program during physical education classes was carried out by several teachers from the same school. These teachers received theoretical and practical instructions for the execution of the program, which consisted of: the theory of safe and non-safe motor responses associated with falls, World Health Organisation data describing the impact of falls on the health of school children, theoretical and practical training on data collection processes and tools (including example videos of subjects performing the test to make sure teachers scored consistently between them), as well as the exercises for each control and intervention session. Each teacher was assigned both intervention and control groups.”
4.- Some things should be fixed regarding how the data was presented in the tables.
4.1.- In table 1 there are some notes without footer explanation, while in the caption there are symbols that are not found in the table.
We thank you for pointing out the inconsistencies in the table. We changed the notes and the abbreviations.
4.1.- Physical education marks are not properly specified: it must be described which assessments have been made, if based on specific sport ability, coordination, agility, or other stuff.
Physical education marks were further detailed as requested on page 2, lines 75-78.
“Parallel to this, for reference purposes, students’ physical education marks obtained in the regular P.E. programme were also recorded. These are given on a 1-10 scale at the P.E. teacher’s discretion, based on the evolution of the student over the academic course on motor skills, as well as theoretical concepts and attitude.”
4.3.- Table 2 compares initial and final tests results for the two groups, but we think it is not clear how they are presented and what are the values in brackets.
We acknowledge the difficulty for the reader to interpret the data in the table; we retitled it and information was added to the footnote to render it more readable.

Reviewer 2 Report
The purpose of the study was to evaluate a falls training program in school-age children through the use of an intervention and control group. The manuscript is moderately well-written, but many grammatical and stylistic corrections are necessary.
There are a few major issues with the manuscript. There is no continuity between the control variables (grade in school, “lifestyle” (see additional comments on this variable), BMI, physical education mark, and sex) and the study itself. The paragraph that explains these results seems like an afterthought and these findings are barely mentioned in the discussion. The discussion in and of itself is very weak, as few other studies have been compared to the present study. The first section of the discussion is simply a restatement of the findings.
A major issue is with the “lifestyle” variable. At one point it is mentioned that this variable is explaining physical activity behaviours, but (as best as I can tell), sport participation is used to define the three groups: sedentary, moderate, and active. There is no mention as to how this variable was measured – were the children polled? Parents? This is certainly not a valid way to measure the concept of “lifestyle” or physical activity, for that matter (as the words sedentary, moderate, and active seem to suggest). If the authors insist on including this variable, it needs to be explained how this variable was measured and the variable needs to be renamed “sport participation” as that is what it is measuring, not lifestyle.
Author Response
Answer to the Reviewer’s comments
Special Issue " Health Promotion and Quality of Life Improvement Throughout the Life Cycle”
of International Journal of Environmental Research and Public Health
Manuscript ID: ijerph-598128
Dear Editor,
Enclosed you will find a revision of our manuscript: Effects of fall training program on automatization of safe motor responses during backwards falls in school-age children. We would like to thank the Editor for giving us the opportunity to resubmit our work and to the reviewers for their thoughtful and constructive comments. We have considered all the suggestions and have incorporated them into the revised manuscript. An itemized point-by-point response to the reviewer’s comments is presented below.
Comments to the Author
Reviewer: 2
1.- The purpose of the study was to evaluate a falls training program in school-age children through the use of an intervention and control group. The manuscript is moderately well-written, but many grammatical and stylistic corrections are necessary.
To avoid grammatical and stylistic errors, we had the manuscript reviewed by a translator and a copywriter.
2.- There are a few major issues with the manuscript. There is no continuity between the control variables (grade in school, “lifestyle” (see additional comments on this variable), BMI, physical education mark, and sex) and the study itself. The paragraph that explains these results seems like an afterthought and these findings are barely mentioned in the discussion. The discussion in and of itself is very weak, as few other studies have been compared to the present study. The first section of the discussion is simply a restatement of the findings.
We acknowledge the continuity issue and have rendered the text more coherent.
Furthermore, we are aware that there is a lack of comparative reasoning in the discussion with external research. The current project focuses on the implementation of a practical program that teaches motor skills to reduce the risk of injury in children when falling. Most research in the field is focused on education and prevention strategies, and although we acknowledge that research in the article and use it to contextualize our findings, we are not able to make a direct comparison. Therefore, we added a section to the limitations paragraph, acknowledging the lack of comparable studies.
3.- A major issue is with the “lifestyle” variable. At one point it is mentioned that this variable is explaining physical activity behaviours, but (as best as I can tell), sport participation is used to define the three groups: sedentary, moderate, and active. There is no mention as to how this variable was measured – were the children polled? Parents? This is certainly not a valid way to measure the concept of “lifestyle” or physical activity, for that matter (as the words sedentary, moderate, and active seem to suggest). If the authors insist on including this variable, it needs to be explained how this variable was measured and the variable needs to be renamed “sport participation” as that is what it is measuring, not lifestyle.
We thank the reviewer for point this out, we agree with this view and variable lifestyle was changed to sport participation to avoid any confusion. Furthermore, sport activity was further defined in page 2, lines 68-74.
“[…]the children’s sport participation was recorded. The latter variable was defined as the amount of sports activity outside school hours and was categorised as low, moderate or high. ‘Low sport participation’ indicated the lack of any sports activities outside school. ‘Moderate sport participation’ was defined as practising sports activities for up to 3 times a week or up to a total of 3 hours per week. ‘High sport participation’ was anything exceeding moderate participation in number of sessions or total time. This information was gathered from the interviews with participants and was combined with the information gathered from parents and teachers.”

Round 2
Reviewer 2 Report
Review 2 for:
Effects of fall training program on automatization of safe motor responses during backwards falls in school-age children
Unfortunately, the vast majority of the comments I made in my original review were not addressed. I have taken the time to go through and update the line numbers. These appear below.
General comment: Be consistent with the use of “P.E.” or “physical education” – both are used interchangeably throughout.
Specific comments:
Abstract:
Line 10/11: Suggest adding “regarding falls” between “prevention and education” and “are not sufficient”
Line 13: Include the mean age of the children included in the study.
Intro:
Line 25: This sentence is making quite a substantial claim and should be referenced.
Lines 25-27: This statement is a bit confusing as you claim that children don’t have fully developed reflexive and balancing abilities, but then make the claim that they develop these abilities during childhood. So which is it?
Lines 29/30: The statement “children who experience medically attended injuries have a higher risk of occurrence” needs to be cited.
Line 31: This sentence/statement seems out of place, and looking at the papers that are referenced, only reference #7 is a suitable reference for this statement.
Line 32: Preventive measures for what?
Line 36: What is meant by “the motor safety skills”? This is an unusual term.
Materials and methods:
Lines 48/49: How many students in each class? What was the total n?
Line 67: “affect the effectivity” does not make sense.
Figure 1: Why was the control group so much smaller than the experimental group?
Line 115: Is this the test used at Baseline If so, this needs to be stated much earlier.
Caption under Figure 2: Why is the explanation under the caption an entire paragraph? This is excessive and much of this information was already explained (verbatim) in the text. Revise significantly.
Results:
Line 179: I assume this is referring to the follow-up test results?
Lines 182-183: It needs to be specified that PHR for the intervention group was lower at the final test, as this is what is reported in Tabel 2. Table 2 does not compare and contrast the experimental group to the control group; thus, the information in this table cannot be used to claim that relative risk in the intervention group was less compared to the control group.
Table 2: This table doesn’t add anything to the article. You’ve already compared and contrasted the control and intervention group in the figure. Eliminate the table and any text referencing it.
Section 3.3: What statistical analysis is this??
Section 3.4: Given that this was not a primary purpose of the study and the way in which these results are presented, this paragraph seems like an afterthought. Suggest removing.
Discussion
Line 228-235: This entire paragraph seems to be beyond the focus of this manuscript. This information also isn’t anything different than what was presented in the results.
Line 238-239: This reference (14) should go after the statement saying that equilibrium exercises were useful in previous studies, not the statement about the present study.
Section 4.2 – the discussion (still) needs more information such as this (e.g., compare and contrast the results of the current study to other studies).
Author Response
Comments to the Author
Reviewer: 2
Unfortunately, the vast majority of the comments I made in my original review were not addressed. I have taken the time to go through and update the line numbers. These appear below.
First of all, we would like to apologize for not meeting the criteria of your previous review. Below the answers, hoping that they are detailed enough. The lines specified here correlate to the document previous to the acceptance of the tracked changes as asked by the editorial board.
General comment: Be consistent with the use of “P.E.” or “physical education” – both are used interchangeably throughout.
The acronym PE is now used throughout the text.
Specific comments:
Abstract:
Line 10/11: Suggest adding “regarding falls” between “prevention and education” and “are not sufficient”
We added “regarding falls” added in line 11.
Line 13: Include the mean age of the children included in the study.
We added the mean age in lines 13-14.
Intro:
Line 25: This sentence is making quite a substantial claim and should be referenced.
We included the reference in line 25.
Lines 25-27: This statement is a bit confusing as you claim that children don’t have fully developed reflexive and balancing abilities, but then make the claim that they develop these abilities during childhood. So which is it?
The sentence has been rewritten to make the meaning clearer.
Lines 29/30: The statement “children who experience medically attended injuries have a higher risk of occurrence” needs to be cited.
We added the reference in line 31.
Line 31: This sentence/statement seems out of place, and looking at the papers that are referenced, only reference #7 is a suitable reference for this statement.
The sentence has been deleted
Line 32: Preventive measures for what?
We modified the sentence to make it clearer. Line 33
Line 36: What is meant by “the motor safety skills”? This is an unusual term.
The sentence has been rewritten to make it clearer. Lines 37 and 38
Materials and methods:
Lines 48/49: How many students in each class? What was the total n?
We added this information in lines 61 and 62.
Figure 1: Why was the control group so much smaller than the experimental group?
The groups were established by stratified random sampling. Control and intervention groups had to share as many characteristics as possible. Therefore, within each year both intervention groups and control groups were formed. Each such group had to be an entire class, since the intervention takes place during the PE classes, this avoids members from control and intervention groups doing the sessions at the same time and place. As each year has three classes, one could choose to have either two control groups and one intervention group or two intervention groups and one control group for each year. The latter option was chosen.
Line 67: “affect the effectivity” does not make sense.
We have rephrased the sentence to make it clearer. Lines 99-100
Line 115: Is this the test used at Baseline If so, this needs to be stated much earlier.
The test is now described from line 76 onwards, at the beginning of the implementation description.
Caption under Figure 2: Why is the explanation under the caption an entire paragraph? This is excessive and much of this information was already explained (verbatim) in the text. Revise significantly.
We have eliminated the redundant caption text under figure 2. Lines 95-96
Results:
Line 179: I assume this is referring to the follow-up test results?
We have substituted this passage, together with table 2, with a comparison of intervention and control group PHR scores. Lines 280-308
Lines 182-183: It needs to be specified that PHR for the intervention group was lower at the final test, as this is what is reported in Tabel 2. Table 2 does not compare and contrast the experimental group to the control group; thus, the information in this table cannot be used to claim that relative risk in the intervention group was less compared to the control group.
See previous answer.
Table 2: This table doesn’t add anything to the article. You’ve already compared and contrasted the control and intervention group in the figure. Eliminate the table and any text referencing it.
We have deleted the table.
Section 3.3: What statistical analysis is this??
We have added the type of analysis (ANCOVA) for the baseline and final test in lines 311-321. Furthermore, we have added the results of this analysis in a table.
Section 3.4: Given that this was not a primary purpose of the study and the way in which these results are presented, this paragraph seems like an afterthought. Suggest removing.
We have deleted the paragraph.
Discussion
Line 228-235: This entire paragraph seems to be beyond the focus of this manuscript. This information also isn’t anything different than what was presented in the results.
We have removed the paragraph in question.
Line 238-239: This reference (14) should go after the statement saying that equilibrium exercises were useful in previous studies, not the statement about the present study.
We have moved the reference to the correct location.
Section 4.2 – the discussion (still) needs more information such as this (e.g., OK compare and contrast the results of the current study to other studies).
We have added comparisons with other studies. Lines 385-390

Round 3
Reviewer 2 Report
My comments have been addressed.